

# Chemical space exploration: how genetic algorithms find the needle in the haystack

Emilie S. Henault, Maria H. Rasmussen and Jan H. Jensen

Department of Chemistry, University of Copenhagen, Copenhagen, Denmark

## ABSTRACT

We explain why search algorithms can find molecules with particular properties in an enormous chemical space (ca $10^{60}$ molecules) by considering only a tiny subset (typically $10^{3-6}$ molecules). Using a very simple example, we show that the number of potential paths that the search algorithms can follow to the target is equally vast. Thus, the probability of randomly finding a molecule that is on one of these paths is quite high and from here a search algorithm can follow the path to the target molecule. A path is defined as a series of molecules that have some non-zero quantifiable similarity (score) with the target molecule and that are increasingly similar to the target molecule. The minimum path length from any point in chemical space to the target corresponds is on the order of 100 steps, where a step is the change of and atom- or bond-type. Thus, a perfect search algorithm should be able to locate a particular molecule in chemical space by screening on the order of 100s of molecules, provided the score changes incrementally. We show that the actual number for a genetic search algorithm is between 100 and several millions, and depending on the target property and its dependence on molecular changes, the molecular representation, and the number of solutions to the search problem.

## INTRODUCTION

Chemical space is the number of possible small organic molecules, which has been estimated to be on the order of $10^{60}$ molecules. Many techniques have been developed to search this chemical space for molecules with desirable properties (*Elton et al., 2019*; *Schwalbe-Koda & Gómez-Bombarelli, 2020*), including genetic algorithms (*Brown et al., 2004*; *Virshup et al., 2013*), variational autoencoders (*Gómez-Bombarelli et al., 2018*; *Winter et al., 2019*), recurrent neural networks (*Segler et al., 2018*; *Brown et al., 2019*; *Sumita et al., 2018*), and generative adversarial networks (*Guimaraes et al., 2017*; *Sanchez-Lengeling et al., 2017*; *Prykhodko et al., 2019*). Rather than screening a user defined library, these methods automatically select a subset of chemical space for screening, usually in an iterative fashion. The size of the subsets typically range between 10,000 and several million, that is a tiny fraction of chemical space yet usually produce good candidates.

$$10^{60} \longrightarrow 10^{3-6} \longrightarrow 1 \tag{1}$$

In this article we discuss how this is possible using genetic algorithms (GAs) as the search algorithm. We use GAs as they are relatively simple and thus easy to

Corresponding author
Jan H. Jensen, jhjensen@chem.ku.dk

interpret, but our general conclusions should also be valid for machine learning-based methods.

The article is organized as follows. First we discuss a related non-chemistry search problem that is conceptually easier to understand but is of roughly similar difficulty: finding a specific sequence of characters. Then we discuss the chemical equivalent, which is finding a specific molecule among the $10^{60}$ possible. Finally, we discuss an example of the more usual molecular discovery problem where there are many solutions.

## COMPUTATIONAL METHODOLOGY

The graph-based GA code used in this study is that described by Jensen (2019) except that, inspired by Brown et al. (2019), elitist selection is used (the next generation is made by combining the highest scoring molecules from the current and previous generation). The GAs use roulette wheel selection to choose parents for mating. The string-based GA code is the same as the graph-based GA code, except for the crossover and mutation operations. For string-based GA the crossover is performed by picking a random cut point between two characters for each parent string and then combining the left and right sub-string from the first and second parent, respectively. In the case of the Shakespeare example described below the same cut point was used for both parents and the fragments are combined so that the children are the same length as the parents. In the case of SMILES and DeepSMILES the local syntax is not considered when choosing the cut-point, for example a cut within [C@H] is allowed. In the case of SELFIES, each unit is enclosed with square brackets, so only cuts between a closing and opening square bracket is allowed. If the crossover does not lead to a valid molecule according to RDKit the process is repeated up to 50 times, after which a new pair of parents are chosen. After crossover the child is mutated at a specified rate (the mutation rate), that is if the mutation rate is 50% then there is a 50% chance that one character in the string is replaced by a randomly chosen character. The allowed characters are those found among 250,000 molecules found in the ZINC data base used in previous studies (Yang et al., 2017; Gómez-Bombarelli et al., 2018; Jensen, 2019) (see Supplemental Information for more information). If the mutation does not lead to a valid molecule according to RDKit the process is repeated up to 50 times, after which the original molecule is returned. In all cases the molecules are Kekulized, meaning that aromaticity is not removed, before mating and mutation operations are applied to increase the chances of making a string that corresponds to a molecule. The string-based molecular representations are not re-canonicalized after mating and mutation operations are applied.

The Tanimoto score used for rediscovery is computed using RDKit (Landrum, 2020) based on ECFP4 circular fingerprints, following Brown et al. (2019). The first excitation energy and associated oscillator strength is computed using the semiempirical sTDA-xTB (Grimme & Bannwarth, 2016) method based on an MMFF94 (Halgren, 1996a, 1996b, 1996c, 1996d; Halgren & Nachbar, 1996) optimized geometry. The geometry is chosen by generating and energy-minimizing twenty random conformations using RDKit and choosing the geometry with the lowest energy.

## RESULTS AND DISCUSSION

### A simple example from Shakespeare

We start by considering a very simple search problem (*Shiffman, 2012*) for which the various factors contributing to successful searches can be demonstrated analytically (see Supplemental Information). The sentence "to be or not to be that is the question" has 39 lower case characters including spaces. It is one of $27^{39} = 6.7 \times 10^{55}$ 39-character sequences, which is roughly the same size as chemical space. Despite this vast search space a simple genetic algorithm (GA) can easily identify the target: using an initial population of 100 randomly generated phrases and a mutation rate of 20% the target phrase is identified after no more than ~300 generations (median ~200), that is the solution is consistently found by evaluating only ca 10,000 to 30,000 sequences out of $6.7 \times 10^{55}$ possible (Fig. 1).

This remarkable feat can be explained as follows: $1-(26/27)^{39}$ or 77% of the $6.7 \times 10^{55}$ possible sequences have at least one correctly placed character compared to the target sequence (Eq. S1). This means that for an initial mating pool of 100 random sequences, an average of 77 sequences will have a score of at least 1. An average $38 \pm 1$ of the 39 positions are correctly represented in at least one gene (Eqs. S4 and S5). Since the score is additive, it is very likely that a crossover will result in a child with a higher score. Indeed simulations show that the tends score increases by 1 with every generation until the score reaches about 20, that is until about half the letters are correctly placed. This makes sense because, on average, each parent contributes half the genetic information and the correctly placed letters are evenly distributed in the initial population. After half the letters and spaces are placed correctly, the score increases more slowly and it can take many generations to place the last character since that tends to occur solely through random mutations in the current GA implementation.

So rather than picturing $6.7 \times 10^{55}$ random sequences that one must sift through, one should picture an enormous number of interconnected paths that connects low-scoring sequences to the target sequence (Fig. 2). Since 77% of the sequences have a score of at least 1, one is very likely to encounter such a path by chance and one can then follow the path to the target sequence using a search algorithm such as a GA. However, such paths only exist if the score increases in a relatively smooth fashion as one gets closer to the target. Figure S1 shows plots similar to Fig. 1, but where there the score only increases if the number of correctly placed characters increases by 2. After about half of the characters are placed correctly, it becomes less likely that a mating operation or mutation increases the score and none of 10 simulations manages to find the correct sequence in 1,000 generations. If the score only increases if the number of correctly placed characters increases by 5, then the GA fails to increase the score beyond 15 (Fig. S1). A chemical example of non-continuous scores is discussed below.

### Rediscovery
#### *String-based approaches*
The closest chemical equivalent to the Shakespeare example described in the previous section is locating a predefined molecule in chemical space, that is rediscovery.

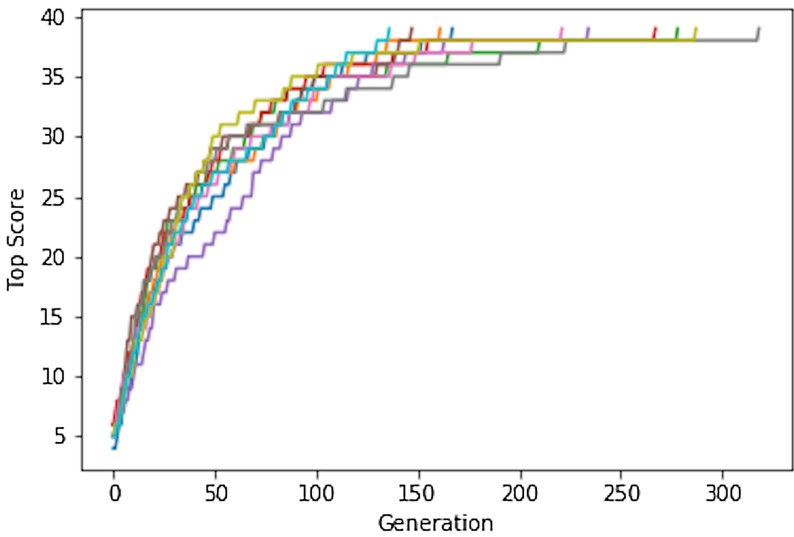

**Figure 1** **The Shakespeare example. Plot of the top score found in the population for each generation for 10 different GA searches.** The population size is 100, so up to 10,000 different sequences are evaluated in 100 generations.

*Brown et al. (2019)* have demonstrated this for three drug molecules: celecoxib, troglitazone, and tiotixene (Fig. 3). Here similarity to the target is measured by the Tanimoto similarity, which is computed by decomposing each molecule into overlapping fragments up to a certain size and then counting how many fragments the two molecules have in common and dividing by the combined total number of fragments. Thus, the Tanimoto score ranges from 0 (no similarity) to 1 (very similar or identical).

Since molecules can be represented as strings (e.g. SMILES strings) we start by using our string-based GA used in the previous example with some minor modifications as described in the Methods section. Otherwise we follow the same procedure as *Brown et al. (2019)*.

The results of 40 SMILES-based GA searches for each of the three molecules are shown in Fig. 4. A total of 45%, 0% and 7% of the searches succeed for celecoxib, troglitazone, and tiotixene, respectively, which all are significantly lower than the 100% success rate for the Shakespeare example. Why do so many of the searches fail? The SMILES strings range in length from 56 to 61 characters and the search uses 25 different characters so the search spaces are larger than in the Shakespeare example. However, the GA used in that example has no problem finding longer sentences (Fig. S2). The other main difference between the SMILES-based rediscovery and the Shakespeare example is the score. In order to compute the Tanimoto score the SMILES string is first converted to a molecular graph, and this conversion fails for a larger portion of the SMILES strings generated using the mating and mutation operations. The failure is primarily due to incorrect SMILES syntax, such as unmatched parentheses or integers denoting ring-closures. Thus, the rediscovery search can only follow paths through sequence space leading to the target molecules that are composed of valid SMILES strings, which is a small subset of all possible

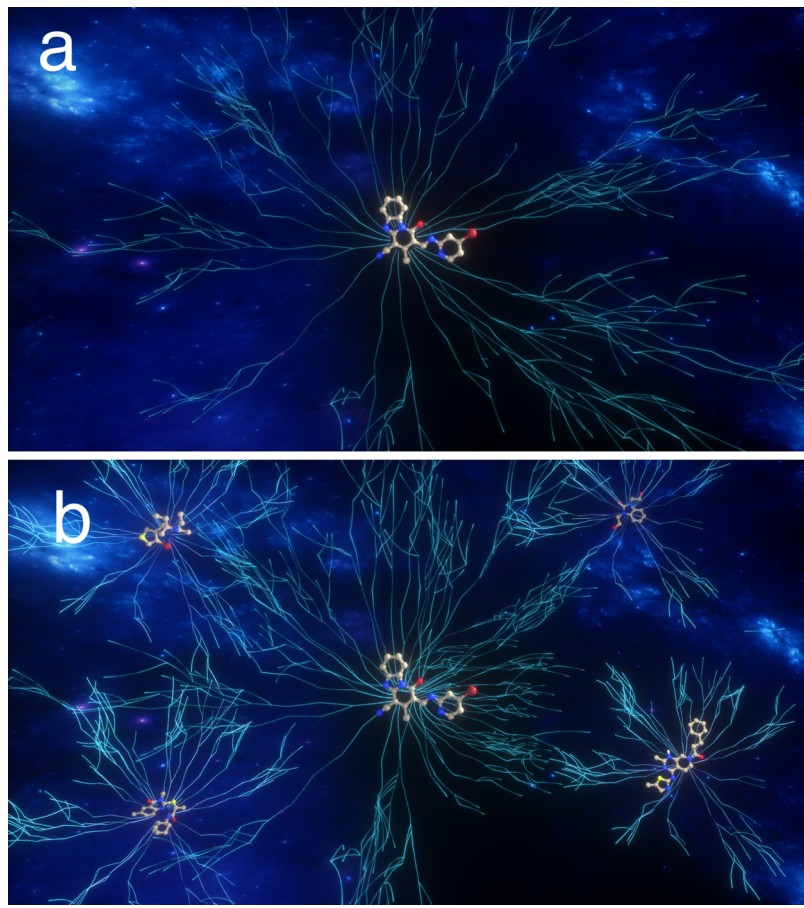

**Figure 2 Pictorial representation of paths (lines) through chemical space leading to target(s), that is molecules with the desired property.** A path connects molecules with non-zero scores and the scores increase incrementally as one gets to the target. In (A) only one molecule has the desired property, while in (B) several molecules have the desired property.

paths (Fig. 2). This makes the rediscovery task intrinsically harder than the Shakespeare example, where the score can be evaluated for nonsensical strings.

Table S1 show the (non-canonical) SMILES strings for the successful SMILES-based GA searches. In the case of troglitazone there were no successful searches, but 15 of the searches resulted in a maximum score of $0.\overline{79}$, which is the second highest score observed, so these string are counted as successful for the current discussion. Many of the SMILES strings show similar patterns. For celecoxib, all but two SMILES strings start with "NS(=O)(=O)C1=.." and end with "..C=C1". For troglitazone all but two SMILES strings start with "CC1=.." and end with "..C1O", or vice versa, and similarly for tiotixene. The most likely explanation is that each respective search starts from the same or similar SMILES strings in the initial population. Indeed inspection of the SMILES strings in the initial population reveal strings with similar patterns (Fig. 5). In the case of celecoxib there are 13 different molecules with the same phenyl-$X$-benzenesulfonamide architecture, which helps explain why celecoxib is rediscovered more frequently than tiotixene,

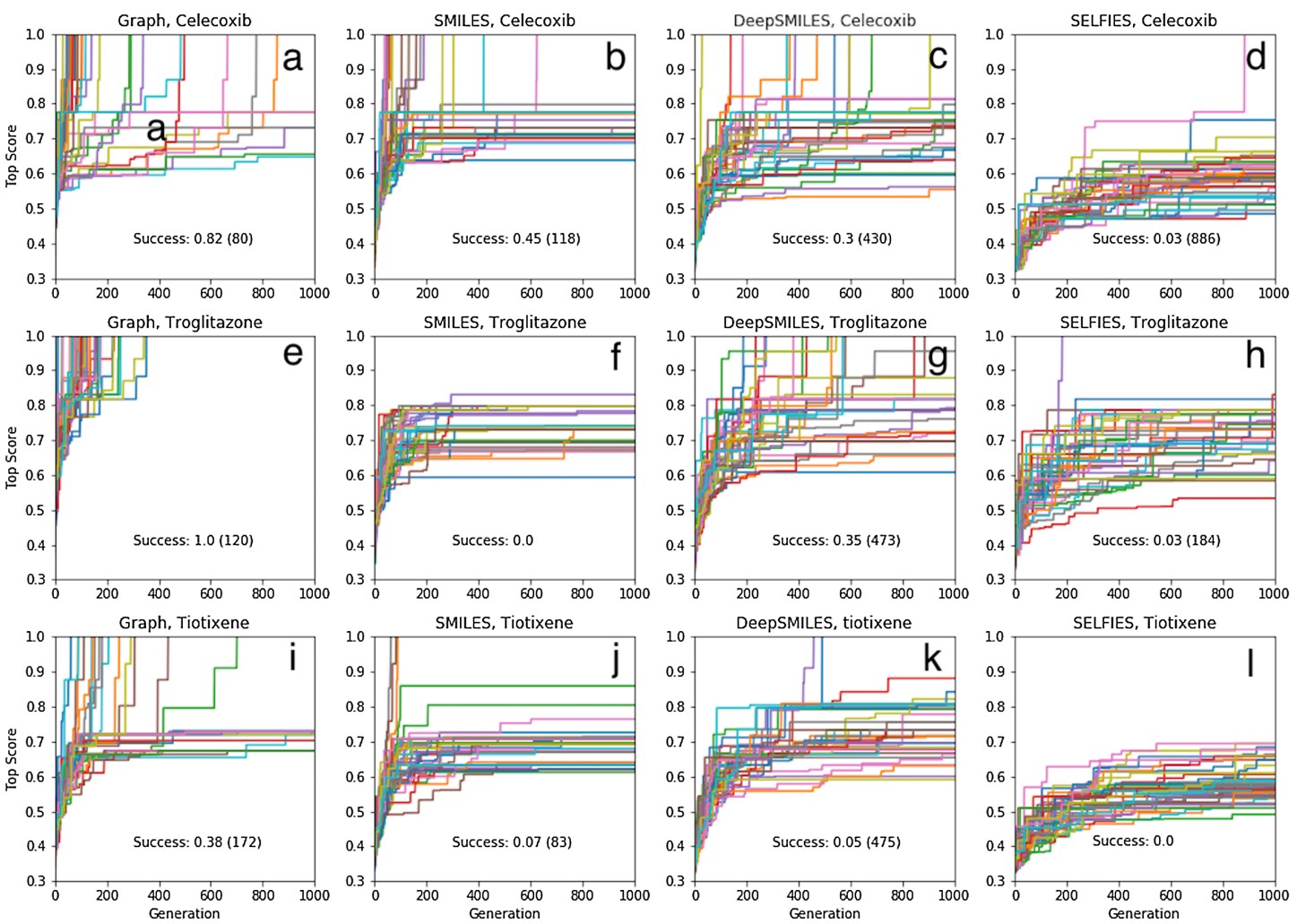

**Figure 3 The three target molecules for rediscovery: (A) celecoxib, (B) troglitazone, and (C) tiotixene.**

**Figure 4 Rediscovery.** Plot of the top score found in the population for each generation for 40 different GA searches for each target molecule (celecoxib (A–D), troglitazone (E–H), and tiotixene (I–L), (Fig. 3) and molecular representation (graph (A, E, I), SMILES (B, F, J), DeepSMILES (C, G, K), and SELFIES (D, H, L)). The score is the Tanimoto similarity to the target molecule computed using ECFP4 circular fingerprints. The population size is 100, so up to 100,000 different molecules are evaluated in 1,000 generations. The mutation rate is 50%. For each plot we show the success rate and the median number of generations for successful runs.

**Figure 5 Examples of SMILES strings obtained by successful SMILES-based GA searches for (A) celecoxib, (B) troglitazone, and (C) tiotixene.** In the case of troglitazone none of the searches were successful, so SMILES with a Tanimoto similarity of 0.79 are shown.

where the SMILES pattern shown in Fig. 5 is the only example in the initial population. In the case of troglitazone, the search has to place a more complicated syntax (COC2=CC=C(CC3SC(=O)NC3=O)C=C2), compared to celecoxib and tiotixene, at the correct position in the string. While this can be done at the 5 position (Fig. 5), it is more difficult at the 2 position (which would result in troglitazone) due to the SMILES syntax of the chromane moiety that is most common in the initial population (Fig. S3). This observation could help explain why none of the SMILES-based troglitazone rediscovery searches are successful.

In the case of troglitazone, the success rate can be improved significantly by using DeepSMILES (*O'Boyle & Dalke, 2018*) (Fig. 4), another string-based molecular representation that doesn't involved matched parenthesis and integers denoting ring-closures. However, using DeepSMILES does not increase the success rate for celecoxib

or tiotixene. Finally, using SELFIES (*Krenn et al., 2019*) does not increase the success rate for any of the three molecules. It is very likely that the performance of the string-based GA searches can be improved significantly by using more sophisticated algorithms (*Bjerrum, 2017*; *Nigam et al., 2019*; *Yoshikawa et al., 2018*). The main point for the purposed of thus study is that the molecular representation is one of the factors that can complicate the exploration of chemical space.

### Graph-based approach

The success rate for rediscovery can be improved significantly by performing the mating and mutation operations directly on the molecule (formally a graph with nodes and edges corresponding to atoms and bonds, respectively) rather than a string representation (Fig. 4). The success rate for tiotixene (38%) is noticeably lower than those for celecoxib (82%) and troglitazone (100%). The reason is that two of the fragments found in tiotixene are not found in the initial population, while the corresponding numbers for celecoxib and troglitazone are zero and one, respectively. The missing fragment for troglitazone relates to the connection between the chromane and benzene group (Fig. S4A). Inspection of the initial population for the troglitazone GA searches shows that it contains several molecules with chromane and anisole groups, which can be combined relatively straightforwardly by a mating operation. The missing fragments for tiotixene relates to the thioxanthene moiety (Fig. S4B). Inspection of the initial population for the tiotixene GA searches shows that the closest match is a single molecule containing a phenothiazine moiety. Constructing the thioxanthene moiety from the molecules in the initial population thus presents a significant challenge and accounts for the lower success rate for tiotixene rediscovery.

At first impression our results indicate that our graph-based GA is able to find a specific molecule in chemical space by evaluating only a very small subset of between ≤35,400 molecules (troglitazone) and ≤1,000,000 molecules (tiotixene). Troglitazone is rediscovered with a 100% certainty in 354 generations or less, where 100 molecules is evaluated for each generation. Tiotixene is rediscovered successfully in 38% of the GA searches, meaning that a minimum of 10 GA searches have to be performed to rediscover tiotixene with a 99% certainty, where each search requires up to 100,000 molecules to be evaluated. However, the initial mating pool was constructed following *Brown et al. (2019)*, that is the 100 top-scoring molecules in a 1.6 million molecule ChEMBL subset, where molecules with an ECFP4 Tanimoto similarity of >0.323 are removed. So constructing the initial population itself requires 1.6 million molecules to be evaluated and this "cost" must be added.

If instead the initial population is constructed as before but from 10,000 molecules chosen from the 1.6 million ChEMBL subset, the success rates are 93%, 70% and 25%, respectively, meaning that at least 2, 4 and 17 GA searches are needed for rediscovery to succeed with >99% certainty (Fig. 6). Thus, between 210,000 and 1,710,000 molecules need to be evaluated to find one particular molecule in chemical space using our GA. All the fragments in celecoxib and troglitazone are in the respective initial population,
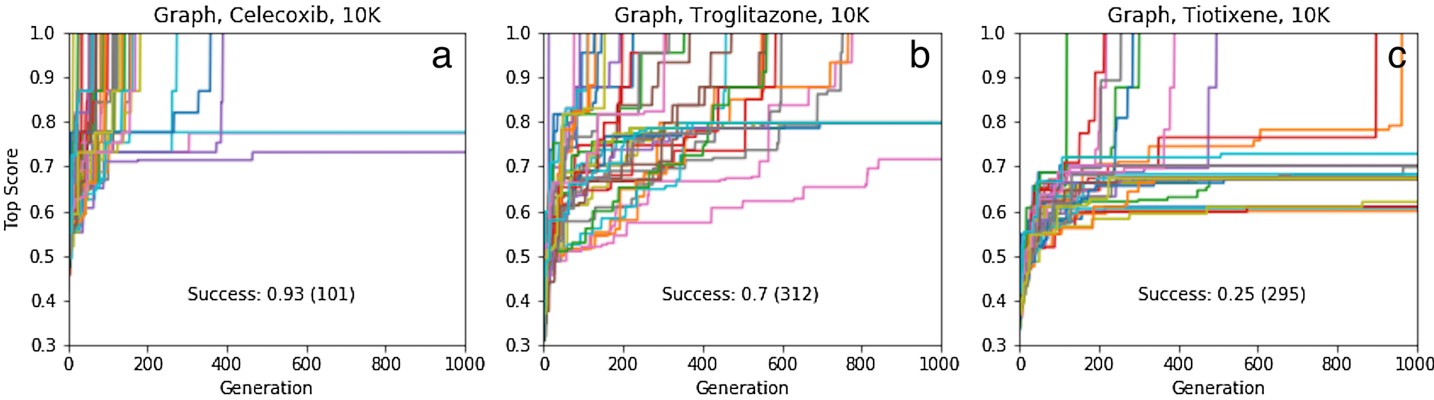

**Figure 6 Same as for Fig. 4 ((A) celecoxib, (B) troglitazone, and (C) tiotixene) but using only the graph based approach and an initial population based on pre-screening 10,000 molecules rather than 1.6 million.**

while two fragments are missing for tiotixene (Fig. S4C). This indicates the substructures that make up the three molecules are relatively common in the ChEMBL data set.

In summary, at least in the case of drug-like molecules it is possible to locate specific molecules in chemical space by evaluating a relatively small subset ($10^{5-6}$). Similarly to the Shakespeare example, the reason is that many of the molecules in chemical space have some structural motifs in common with the target molecule. Search algorithms like GAs can then combine these structural motifs to create molecules that are increasingly similar to the target molecules. The order in which these fragments are combined correspond to different (interconnected) paths in chemical space that all lead to the target molecules (Fig. 2). There are a vast number of such paths, so it is highly likely to randomly encounter at least one such path, which can then be followed to the target. In this particular case, the search is frustrated by the use of the Tanimoto similarity as the scoring function, since it is only a semi-continuous function (cf. Fig. S1) of the molecular structure in the sense that *all* atoms and bonds in a fragment must be placed correctly before the fragment is counted as found.

## Absorbance

Rediscovery is mainly interesting because there is only one solution (or very few solutions) and thus serves to test the limits of chemical space search algorithms. Most target properties will have several solutions in chemical space and may thus be easier to find. To illustrate this, we search for molecules that absorb light at 200, 400, and 600 nm with an oscillator strength ($\omega$) of $\geq 0.3$. The score is given by

$$\text{Score} = e^{-\frac{1}{2}\left(\frac{\lambda - \lambda_t}{\sigma}\right)^2} + \min(\omega, 0.3)/0.3 \tag{2}$$

where $\lambda$ is the computed absorption wavelength of the molecule, $\lambda_t$ is the target wavelength, and $\sigma$ is 50 nm. The GA searches are terminated if the top score in the population is within 0.01 of the maximum possible score of 2.0. The absorption wavelength and oscillator strength is computed using the xTB-sTDA method (*Grimme & Bannwarth, 2016*) based on a low-energy molecular structure computed using the
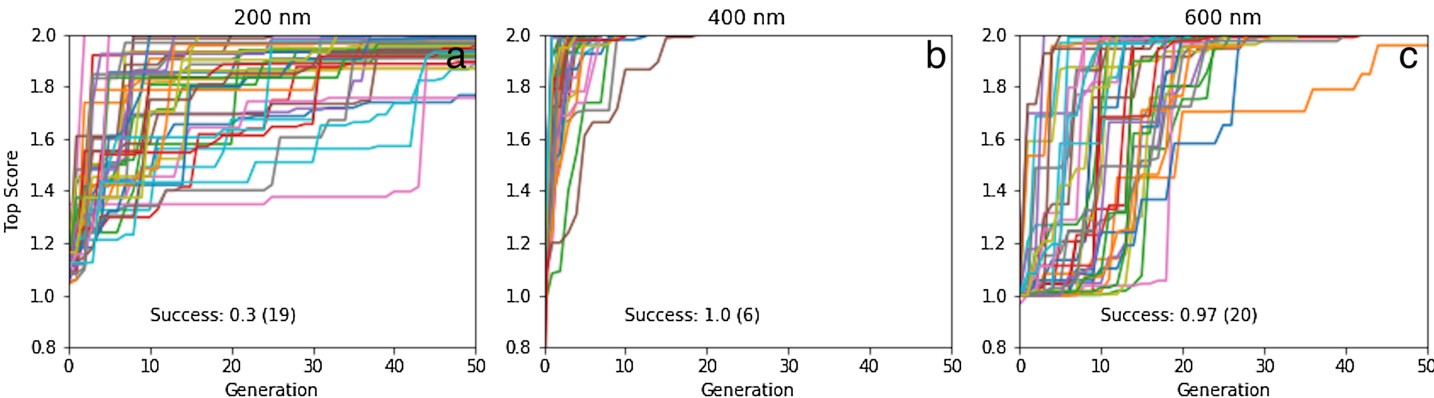

**Figure 7** Plot of the top score found in the population for each generation for 40 different GA searches for molecules that absorb at (A) 200, (B) 400, and (C) 600 nm, all using a graph-based molecular representation. The population size is 20, so up to 1000 different molecules are evaluated in 50 generations. The mutation rate is 5%. For each plot we show the success rate and the median number of generations for successful runs.

MMFF94 force field as implemented in RDKit. The low-energy structure is computed by generating and energy-minimising 20 random conformations using RDKit and choosing the conformer with the lowest MMFF94 energy. The molecules for the initial population are chosen randomly from the first 1,000 molecules in 250,000-molecule subset of the ZINC data base that we have used previously. *Jensen (2019)* Molecules that absorbed within 100 nm of the target wavelength were excluded from the initial population. The goal of these simulations is to illustrate the use of GAs with a scoring function that has a complex dependence on the molecular structure and a target property with multiple solutions, not to find stable, synthetically accessible molecules for experimental testing.

The results for 40 50-generation GA searches with a population size of 20 are shown in Fig. 7. The success rates are 100% and 97% for 400 and 600 nm, while only 30% for 200 nm. For 400 and 600 nm, the median number of generations needed to find to find a molecule with the target property is 6 and 20 generations, respectively, which corresponds to screening only 120 and 400 different molecules. While the success rate is comparatively low for 200 nm (requiring up to 13,000 molecule evaluations) it is still impressive given the small population size, initially constructed from randomly chosen molecules (i.e. no pre-screening like for rediscovery). Inspection of the molecules (Fig. 8; Figs. S5–S7) show that, as expected, they are all different. Thus, there are many molecules in chemical space that satisfy the search criterion with, presumably, many different paths leading to each target, as shown for rediscovery, which increases the chances of success (Fig. 2).

For 200 nm, all but two GA searches achieved a score of >1.85, which corresponds to wavelength with 28.5 nm of the target value (assuming the oscillator strength is >0.3). So the majority of the searches get reasonably close to the target, but fail to reach the success criterion of 1.99, which corresponds to a wavelength within 7 nm of the target value. The most likely explanation is that it requires a larger change in excitation energy to shift low wavelength excitations. For example, a shift from 207 to 200 nm requires a change

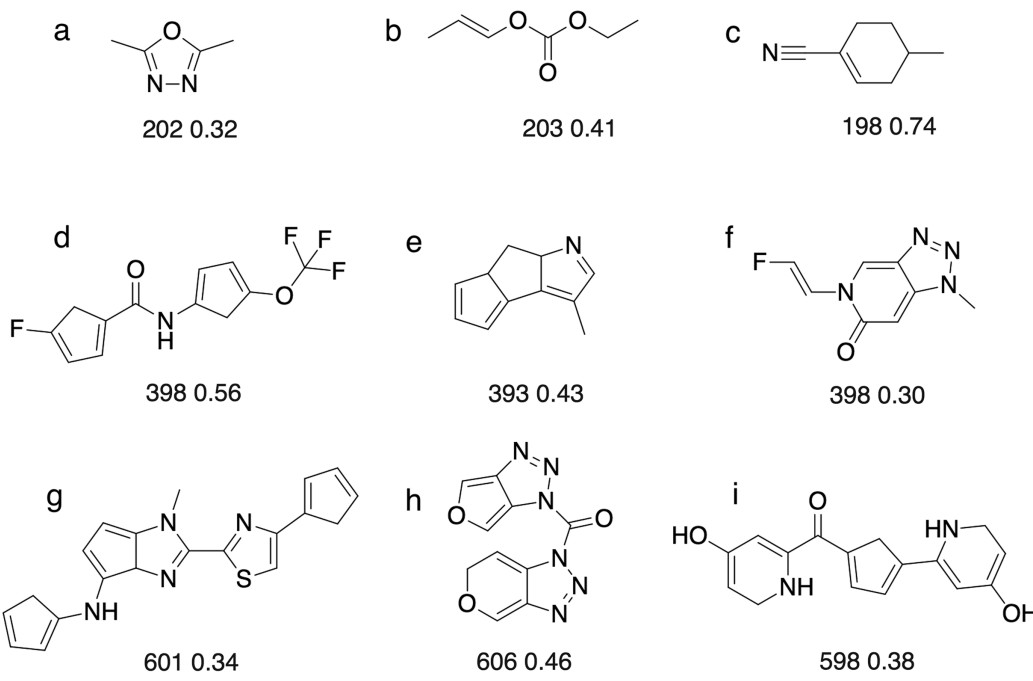

**Figure 8 Some of the molecules ((A)–(I)) found by the GA searches for molecules that absorb at a certain wavelength.** Below each molecules is the computed absorption wavelength (in nm) and oscillator strength. We recognise that some of these molecules may not be stable (e.g. cyclopentadiene groups tend to dimerise) or represent the most stable tautomer. We merely use absorbance as a score that has a complex dependence on the molecular structure.     

in excitation energy of 0.21 eV, compared to 0.05 eV (ca 1 kcal/mol) for a shift from 407 to 400 nm. Thus, absorption wavelengths around 400 nm are easier to fine-tune using relatively modest molecular changes, compared to 200 nm. Conversely, in the case of 600 nm, almost any change to the structure can easily change the excitation wavelength by 7 nm, so it becomes more difficult to hit the target wavelength exactly compared to 400 nm. This underscores the importance of smooth, incremental scoring functions for search efficiency (Fig. S1).

## CONCLUSIONS

This paper explains how search algorithms can find particular molecules in an enormous chemical space ($10^{60}$ molecules) by considering only a tiny subset (typically $10^{3-6}$ molecules). We use a simple, non-chemistry related search problem that is easy to interpret quantitatively. We show that a genetic algorithm (GA) can find one particular 39-character sequence by considering at most 30,000 out of $6.7 \times 10^{55}$ possible sequences (Fig. 1). The reason is that 77% of the $6.7 \times 10^{55}$ possible sequences have at least one correctly placed character (with a score of 1), so it is easy to find such sequences by random chance. Search algorithms like GAs then combine these sequences to make higher-scoring sequences, in an iterative fashion, until the target sequence is obtained. If we view closely related sequences with correctly placed characters as being "connected" then we can envisage the search space as being filled with an enormous number of interconnected paths

that connect sequences with few correctly placed characters to the target sequence (Fig. 2). It is easy to find a distant point on one of these paths and relatively easy to follow the path to the target using search methods such as GAs, provided the score changes incrementally as the sequence is changed (Fig. S1). As step along the path represents an edit of the sequence so the length of a given path from a given point to the target is the so-called edit distance, where the change in a single character corresponds to an edit distance of one. This means that the shortest possible path from a sequence with only one correctly placed character to the target is only 38. So while the sequence space is vast, the shortest distance between any pair of points involves at most 39 changes.

The closest chemical equivalent to the simple string search example is locating a predefined molecule in chemical space, that is rediscovery. Rediscovery, using text strings (SMILES, DeepSMILES and SELFIES) to represent the molecules, is shown to be significantly more challenging even though the mechanics of the GA search (i.e. the mating and mutation operations) are very similar. Most string-based searches fail to find the target after 100,000 molecule evaluations starting from initial populations made by pre-screening over 1.6 million molecules (Fig. 4). The primary difference between the simple phrase search and rediscovery is that in the latter case the score can only be evaluated for strings that correspond to valid molecules, while in the former case all strings can be scored. Since most string-based mating a mutation operations lead to strings with invalid syntax and zero scores for the rediscovery search, there are many fewer paths leading to the target (Fig. 2) compared to the simple string search.

In one of the the three rediscovery targets (troglitazone) the success rate can be improved by using DeepSMILES, a string-based molecular representation with simpler syntax compared to SMILES. It is also quite likely that the success can be improved further by more sophisticated mating a mutation operations designed specifically for particular syntax associated with each molecular representation. The success rate can be improved by performing mating and mutation operations directly in the molecular graph (i.e. the atom and bonds, Figs. 4 and 6), where a particular molecule can be rediscovered with >99% certainty by evaluating between 210,000 and 1.7 million ($10^{5-6}$) molecules—a very small fraction of chemical space. In analogy with the simple phrase example, the reason is that the chemical "alphabet" of organic chemistry is relatively small ca ten different atoms and three different bonds. So it is quite likely that a randomly chosen molecule has *something* in common (e.g. a C–C bond or a pyridine ring) with the target molecule and thus lies on a path that a search algorithm can follow to the target (Fig. 2). For example, more than 99.9% of the 1.6 million molecules in the ChEMBL data set, used to construct the initial populations, have a non-zero Tanimoto similarity with the three rediscovery targets (Fig. 3).

Most drug-like molecules have at most 50 atoms and bonds, so the number of changes needed to inter-convert two very different molecules (the so called graph edit distance) is generally less than 100. So, while chemical space is vast, the ideal search algorithm can traverse it very quickly—as long as the desired property changes incrementally.

While rediscovering one molecule in chemical space can require the screening of $10^{5-6}$ molecules, finding a molecule with a particular property, such as absorbance at a particular wavelength, can often be accomplished more efficiently since there tends to be many different molecules with the desired property (Figs. 2B, 7 and 8). For example, finding molecules that absorb at 200 ± 7, 400 ± 7 and 600 ± 7 nm requires the screening of up to 13,000, 120 and 400 different molecules.

This study focuses on GAs as they are relatively simple and thus easy to interpret, but our general conclusions should also be valid for other generative models aimed at searching for molecules with specific properties. Such generative models typically combine a machine-learned molecular representation with a standard search algorithm such as swarm optimization (*Winter et al., 2019*), hill climb (*Brown et al., 2019*), or Monte Carlo tree search (*Sumita et al., 2018*). Like most search algorithms, these algorithms are designed to find and incrementally follow paths through search space towards the desired goal (Fig. 2), similarly to GAs.

### Funding
Maria H. Rasmussen is supported by a research grant (00022896) from VILLUM FONDEN. The funders had no role in study design, data collection and analysis, decision to publish, or preparation of the manuscript.

### Grant Disclosures
The following grant information was disclosed by the authors:
VILLUM FONDEN: 00022896.

### Competing Interests
Jan H. Jensen is an Academic Editor for PeerJ.

### Author Contributions
- Emilie S. Henault conceived and designed the experiments, performed the experiments, analyzed the data, performed the computation work, authored or reviewed drafts of the paper, and approved the final draft.
- Maria H. Rasmussen conceived and designed the experiments, performed the experiments, authored or reviewed drafts of the paper, and approved the final draft.
- Jan H. Jensen conceived and designed the experiments, performed the experiments, analyzed the data, performed the computation work, prepared figures and/or tables, authored or reviewed drafts of the paper, and approved the final draft.

### Data Availability
The GA code for the Shakespeare example, input and output files, and data analysis code is available at GitHub: https://github.com/jensengroup/GA_ChemSpace_exploration/tree/v0.0.1.

The graph-based GA code is available at GitHub: https://github.com/jensengroup/GB-GA/releases/tag/v1.0.

The string-based GA code is available at GitHub:
https://github.com/jensengroup/String-GA/releases/tag/v0.0.

## Supplemental Information

Supplemental information for this article can be found online at http://dx.doi.org/10.7717/peerj-pchem.11#supplemental-information.

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
