# Peer review of "Chemical space exploration: how genetic algorithms find the needle in the haystack"

_PeerJ Physical Chemistry, doi:10.7717/peerj-pchem.11_

## Round 0.1 · original submission · Minor Revisions

As noted by reviewer 1, the list of references is quite short. Given that molecular design and global optimization have been very active fields in computational chemistry for the last 20 years, some additional context of how the presented results relate to previous research would be appropriate.

Please also address the other comments raised by both reviewers.

Reviewer 1 ·

Basic reporting

This paper cites only 18 references even though it addresses a rather 'classic' problem using rather 'classic' methods. 6 of the 18 references correspond to potential energy methods which are not the point of this study. Comparison to performance of other optimization runs published in design papers in the literature is lacking entirely. As such, I am skeptical about the scholarship level of this manuscript. For example, Geerlings and co-workers published some best first search molecular design results a couple of years ago where they also found surprisingly rapid convergence in chemical space. The authors offer little to no context regarding this and other related work in the field. All other boxes (English/Structure/self-contained) check.

Experimental design

I see two problems with this manuscript. First, the research question is, to the best of my understanding, not entirely well defined. Typically, when performing molecular design, some target property value should be given, and the optimisation algorithm has to search chemical space in order to minimise deviation from target property. This is what the authors do for the absorbance. It is less clear to the reader what is the actually target in their 'Graph-based approach' section. If it is just the Tanimoto similarity I do not really understand the point since the target is then known already and a simple greedy best first search algorithm should trivially lead to the result in (practically) no time. Or is this interesting because only a GA should be used? In any case, I do not think that this is sufficiently well explained.
Secondly, the example target values for the absorbance (200, 400, and 600 nm) seem ad hoc, and it is hard to decipher for the reader how 'hard' they really are. If the absorbance distribution of their molecular space had, for example, peaks at those three values, it should be relatively easy to rapidly find examples that get close. As such, it is not clear how such a hidden bias might have affected the results and the conclusions drawn. I think that a distribution plot of some representative sub-sample should be shown, and the authors should also include target values which lie outside of that distribution, in order to see if their conclusions also hold for the extremes.
All other boxes (primary research/rigorous investigation/methods) check.

Validity of the findings

The conclusions should be adapted in response to the changes requested before. All other boxes check.

Additional comments

This is an interesting study with interesting findings. There are some points which should be adressed, but I think that it can be easily done. It might also be worthwhile to include some speculations in how far the conclusions depend on the choice of GA as an optimization algorithm, and how they could change if other algorithms were used.

Reviewer 2 ·

Basic reporting

The authors use precise English and the professional words. However, some mistakes still exist and the authors should proofread the paper carefully. The background may lack some proper description.

Experimental design

The problems the paper aims to tackle are clearly stated.

Validity of the findings

In this manuscript, the authors describe the genetic algorithms to find particular molecules in an enormous chemical space by considering only a tiny subset. The method has been proved exactly in authors’ previous work. Here, the rediscovery of specific molecules is discussed by strings-based and graph-based approach and the search for the target properties is also analyzed in detail. The explanation about particular molecules can be easier found by search algorithms.

Additional comments

(1) The mutation rates used in string-based and graph-based genetic algorithms should be specified in manuscript.
(2) It would be better to demonstrate that whether the search for molecules with target properties depends on the properties of initial population or not.
(3) For each generation of different searches, the scores have different degrees of improvement. It would be better to show the processes of molecules changes.

---

## Round 0.2 · accepted · Accept

The changes adequately address the comments of the reviewers.